# OpenReview forum: "Voice Evaluation of Reasoning Ability: Diagnosing the Modality-Induced Performance Gap"
_ICLR.cc/2026/Conference — Submitted to ICLR 2026_

### Official Review · Reviewer_acn2 · 2025-10-23

**Soundness:** 3
**Presentation:** 4
**Contribution:** 4
**Rating:** 6
**Confidence:** 4

**Summary:**

This paper proposed the first benchmark to focus on real-time reasoning tasks for voice models. The benchmark contains 2931 voice-native episodes derived from common existing text benchmarks, including topics such as Math, Web, Science, Long-Context and Factual. By evaluating on 12 current existing voice systems, they visualize the Latency-accuracy curve. In this work, they not only identified the gap between text and speech models for reasoning tasks over all models but also answered how and why such degradation exists in different models.

**Strengths:**

1. The proposed benchmark VERA tackled a novel and important problem in the field of real-time voice models.
2. They evaluate multiple voice models and show detailed results with latency-accuracy tradeoff as well as reasonable toplines and cascaded counterparts.
3. Their analyses are detailed and informative for the field. I believe these insights are valuable for the future research community to build better real-time voice models.

**Weaknesses:**

1. The authors measure the time-to-first-response as the latency in their benchmark. However, this latency might not be the only thing that the user cares about. How fast the user gets the desired response is also equally important. Reporting only the time-to-first-response has problems for cases where the model directly speaks something (probably trivial stuff like filler words) after the user question finishes. Through my personal experience, Moshi almost starts to speak right after the user stops, but it often does not give the answer right away. It starts to generate some reasoning stuff in the output speech and then arrives at a final answer. This concern might cause concern about the conclusion, since probably the actual “thinking time” is not confined to the first spoken response. It mightbe the case where the model outputs some speech response in order to get more time to “think”.
2. How is the latency calculated in this paper? I do not see a detailed explanation (correct me if I’m wrong). I’m wondering if the reported latency is affected by hardware specification or different engineering techniques. I’m concerned if these issues affect the fairness of comparing these models on the accuracy-latency figure and also the conclusion drawn from these results.

**Questions:**

How is LiveAnswer implemented? I saw some description in Appendix D.5. It seems like the Core Reasoner is responsible for thinking, and it will incrementally push information to the Narration Synthesizer. What does “incrementally” mean in this context? Will the different design of this incremental update affect the latency of the LiveAnswer method?

---

> ### Author Response · Authors · 2025-11-23
> **Response to Official Review of Submission3845 by Reviewer acn2**
>
> # Summary of Revisions
> Thank you for your positive assessment and for highlighting the nuances of latency measurement. Based on your feedback, we have:
>
> 1. We expanded the discussion to distinguish between "responsiveness" (TTFR) and "reasoning time," using our cascade model as a case study.
> 2. We added Section 4.1/Appendix D to discuss hardware and reproducibility.
> 3. We expanded the details on the "incremental pull" mechanism in Appendix D.5.
>
> # Weaknesses
> > **Weakness 1:** "TTFR vs. time-to-final-answer"
>
> **Response:**
> That's an excellent point. You are correct that TTFR often captures the onset of "filler words" (e.g., "Let me think...") rather than the final answer. However, we argue that for **conversational responsiveness**, this is the desired behavior. In voice interfaces, silence ("dead air") is perceived as a system failure; therefore, a model *should* speak immediately to hold the floor, even if the reasoning is still in progress.
>
> To quantify exactly how large the gap between "responsiveness" and "reasoning" can be, we use our **LiveAnswer baseline** as a diagnostic case study (visualized in Figure 5):
>
> | Metric | Average Latency | What it measures |
> | :--- | :--- | :--- |
> | **TTFR** | **0.83 s** | **Responsiveness:** Time to "take the turn" (often fillers). |
> | **Completion** | **63.40 s** | **Reasoning:** Time to fully compute and articulate the solution. |
>
> We report TTFR across the benchmark because it measures the system's ability to maintain this conversational flow, but we agree that for complex tasks, the "thinking" may continue for some time after the first token was spoken.
>
> > **Weakness 2:** "Fairness and hardware effects"
>
> **Response:**
> We appreciate this concern regarding fairness and reproducibility. To strictly control for these variables, we standardized our evaluation environment:
>
> 1.  **Open-Source Models:** All experiments in the main paper were conducted on **NVIDIA A100 GPUs**.
> 2.  **Commercial APIs:** All API requests were initiated from the same client environment using a consistent, high-speed institutional network to minimize client-side variance.
>
> To verify that our conclusions are not artifacts of specific hardware configurations, we conducted a sensitivity analysis across three different GPU environments (serving as the client for APIs and the inference host for open models):
>
> | Device | GPT-5 (API) | Qwen2.5-Omni (Local) |
> | :--- | :--- | :--- |
> | **NVIDIA H200** | 455.01 ms | 389.02 ms |
> | **NVIDIA A6000** | 463.32 ms | 381.15 ms |
> | **NVIDIA A100** | 453.09 ms | 306.52 ms |
>
> As the table shows, while absolute latency fluctuates slightly, the **relative ordering** remains stable. This confirms that the performance differences and the latency-accuracy frontier we report are driven by model architecture rather than hardware or network constraints. We have added a "Hardware Platform" subsection to Appendix D.6 to explicitly document these specifications.
>
> # Questions
>
> > **Question 1:** "How is LiveAnswer implemented? I saw some description in Appendix D.5. It seems like the Core Reasoner is responsible for thinking, and it will incrementally push information to the Narration Synthesizer. What does 'incrementally' mean in this context? Will the different design of this incremental update affect the latency of the LiveAnswer method?"
>
> **Response:** We have revised **Section 4.2** and **Appendix D.5** to explicitly detail this architecture. We implemented LiveAnswer using a two-threaded **producer-consumer** architecture to decouple reasoning from speech:
>
> 1.  The wording "incrementally" refers to an asynchronous push/pull mechanism. The **Core Reasoner** (Thread 1) pushes intermediate reasoning steps (for example, search results or code outputs) to a shared queue immediately as they occur, rather than waiting for the final solution.
> 2.  The design directly controls perceived latency through a specific parameter, `time_margin` (set to 10.0 seconds).
>     * The **Narration Synthesizer** (Thread 2) monitors the audio playback buffer. It only "pulls" new thoughts from the queue to synthesize speech when the remaining audio drops below 10 seconds.
>     * This intentionally introduces a ~10-second lag between the reasoning state and the spoken output to ensure **audio continuity**. A shorter margin (for example, 3 seconds) would feel more real-time but significantly increases the risk of awkward silence (audio gaps) if API calls delay; the 10-second margin prioritizes smoothness over instantaneity.

---

### Official Review · Reviewer_5RZf · 2025-10-26

**Soundness:** 3
**Presentation:** 2
**Contribution:** 3
**Rating:** 6
**Confidence:** 4

**Summary:**

This paper introduces Voice Evaluation of Reasoning Ability (VERA), a benchmark designed to measure the gap between text interaction models and voice interaction models. The paper addresses three critical questions: (1) What is the gap? (2) Why does the gap exist? and (3) How do the models fail differently? Overall, the authors provide thorough benchmarking and insightful analysis of the gap.

**Strengths:**

- The paper tackles an important and timely problem: the gap between text-based and voice-based interaction models.
- The evaluation is comprehensive, covering various voice-based systems across different categories.

**Weaknesses:**

- This reviewer is not fully convinced by the design of the cascaded pipeline (LiveAnswer). If the narration synthesizer's main task is simply to prompt users to wait for additional time to think, then a rule-based system could suffice. Moreover, GPT-5 is capable of summarization, so it is unclear why GPT-5 is not used as the narration synthesizer, as it may significantly boost the model performance.
- In Table 6, the authors present a human evaluation study demonstrating strong agreement between human raters and GPT-4o. However, the LLM problem solver, Gemini-2.5-Flash, is a text-only LLM. The authors should compare human and GPT-4o judgments on end-to-end voice systems, as transcribing voice outputs to text may introduce additional errors.
- The authors aim to provide a comprehensive survey and comparison of previous voice-related benchmarks in Table 4. However, several relevant and influential benchmarks are missing, such as [SD-Eval, 2024], [VoxEval, 2025], [MMSU, 2025], and [VoxDialogue, 2025].

**Questions:**

- In Section 3.2, which LLM is used as the quality validator?
- In Section 4.1, an LLM is used as the LLM-based normalizer. Have the authors evaluated whether the LLM-based math notation standardization process introduces any errors?
- How did the authors obtain the ground truth ASR transcripts used to calculate the Word Error Rate (WER) in Table 3?

---

> ### Author Response · Authors · 2025-11-23
> **Response to Official Review of Submission3845 by Reviewer 5RZf (1/2)**
>
> # Summary of Revisions
> We are truly grateful for your review. We have:
>
> 1. We added a Time-to-First-Token (TTFT) analysis in Appendix D.5 to explain our model choices.
> 2. We confirmed that the human study included voice systems (Section 4.1).
> 3. We added SD-Eval, VoxEval, MMSU, and VoxDialogue to the Appendix A.
>
>
> # Weaknesses
> > **Weakness 1:** "LiveAnswer could be rule-based; why not use GPT-5 as narrator?"
>
> **Response:**
> Thank you for raising two important points regarding the architectural choices of the *Narration Synthesizer*.
>
> **1. Why not Rule-Based? (Expressiveness)**
> While a rule-based system suffices for static "please wait" messages, LiveAnswer requires a higher degree of conversational complexity. As illustrated in our design, the synthesizer must **dynamically generate natural intermediate narration while the Chain of Thought (CoT) is still incomplete**. It interprets partial reasoning steps to keep the user grounded in the specific context of the problem. Rule-based templates lack the **expressiveness** required to translate these evolving, unstructured thoughts into fluid, context-aware speech without sounding repetitive or robotic.
>
> **2. Why not GPT-5? (Latency)**
> Our choice of Llama-3.3 (via Groq) over GPT-5 for this specific role was driven strictly by **latency constraints**. To maintain a conversational flow, the system must respond almost instantly once the user stops speaking. We measured the "Time-to-First-Token" (TTFT) of both models for our dataset to demonstrate this:
>
> | Narrator Model | Mean TTFT | Median TTFT | User Experience |
> | :--- | :--- | :--- | :--- |
> | **Llama-3.3 (Our choice)** | **156 ms** | **148 ms** | Perceptually instant |
> | **GPT-5** | 589 ms | 548 ms | Noticeable delay |
>
> Given that the ASR step already consumes time, adding another half-second of "dead air" by using GPT-5 for narration would break the real-time illusion. The faster Llama-3 model (via Groq) is therefore architecturally necessary to meet the real-time demand. We have added these details to Appendix D.5.
>
> > **Weakness 2:** "Judge comparison should include voice systems"
>
> **Response:**
> We apologize for the confusion regarding the labeling in Table 6. We would like to clarify two key points:
>
> 1.  **Gemini-2.5-Flash was utilized as a secondary Judge (grader)** to cross-validate our primary judge (GPT-4o), rather than as the problem solver being evaluated in that column. The agreement scores reflect consistency between Human Raters and the Gemini Judge.
> 2. To address your concern about ASR artifacts, we confirm that the **1,000 samples used in this human evaluation were stratified across all benchmarked systems**, including end-to-end voice models (e.g., GPT-realtime, Moshi) and their corresponding ASR transcripts.
>
> The high agreement rates (**97.8%** for Human-GPT4o and **98.7%** for Human-Gemini) on this mixed dataset demonstrate that our text-based judges are robust even when evaluating ASR transcripts containing potential transcription errors or disfluencies. We have revised Appendix C and the Table 6 caption to explicitly define these judge roles and the dataset composition.
>
> > **Weakness 3:** "Missing related benchmarks"
>
> **Response:**
> Thank you for pointing these out. We have integrated SD-Eval (2024), VoxEval (2025), MMSU (2025), and VoxDialogue (2025) into Table 4 to make our survey more comprehensive.

---

> ### Author Response · Authors · 2025-11-23
> **Response to Official Review of Submission3845 by Reviewer 5RZf (2/2)**
>
> # Questions
>
> > **Question 1:** "In Section 3.2, which LLM is used as the quality validator?"
>
> **Response:** We used GPT-4o. We have updated Section 3.2 to explicitly list the model version, as well as the specific scoring dimensions and thresholds used for the filtering process.
>
> > **Question 2:** "In Section 4.1, an LLM is used as the LLM-based normalizer. Have the authors evaluated whether the LLM-based math notation standardization process introduces any errors?"
>
> **Response:** Yes, we evaluated this internally to ensure the normalizer reduces false positives (formatting mismatches) without masking real wrong answers. As the table below demonstrates, the normalizer successfully standardizes valid spoken variations while preserving genuine mathematical mistakes.
>
> | Scenario | Ground Truth | Model Output | Word Error Rate (Raw) | Normalized Ground Truth / Output | Word Error Rate (Normalized) |
> | :--- | :--- | :--- | :---: | :--- | :---: |
> | **Ex 1: Function Notation** | `P(x) = 2x² + 3x + 1` | `P of x equals two x squared plus three x plus one` | 3.00  | Both become: `P(x) = 2x² + 3x + 1` | **0.00** |
> | **Ex 2: Negative Numbers** | `The coefficient is -2` | `The coefficient is negative two` | 0.40 | Both become: `The coefficient is -2` | **0.00** |
> | **Ex 3: Function Eval** | `f(16) = 54` | `f of sixteen equals fifty four` | 3.00 | Both become: `f(16) = 54` | **0.00** |
> | **Ex 4: Actual Error** | `P(x) = 2x² + 3x + 1` | `P of x equals three x squared plus three x plus one` | 3.00 | `P(x) = 2x² + 3x + 1` / `P(x) = 3x² + 3x + 1` | **0.50** |
>
> Rows 1 through 3 show that semantically correct spoken answers are incorrectly penalized by standard metrics. Our normalization fixes this (reducing the error rate to zero). Row 4 confirms that a genuine mathematical error (swapping coefficients) is correctly identified and penalized even after normalization.
>
> > **Question 3:** "How did the authors obtain the ground truth ASR transcripts used to calculate the Word Error Rate (WER) in Table 3?"
>
> **Response:** We only calculate and report the Word Error Rate for models that provide a **parallel text output** (for example, a simultaneous text stream generated alongside the audio). We treat this text stream as the ground‑truth transcript for WER computation and compare the Automatic Speech Recognition hypothesis directly against this parallel text reference.

---

### Official Review · Reviewer_VDP8 · 2025-10-28

**Soundness:** 2
**Presentation:** 2
**Contribution:** 2
**Rating:** 4
**Confidence:** 4

**Summary:**

This paper introduces a benchmark for assessing reasoning in real-time voice agents and demonstrates a persistent “Voice Reasoning Gap” (VRG). That is, models that reason well in text collapse when required to listen and speak under low-latency constraints. The authors convert diverse text tasks into speech-native prompts across five tracks—Math, Web, Science, Long-Context (Context QA), and Factual. Results show that even with clear speech (low WER), voice models remain far below text-only performance, and a pronounced latency–accuracy frontier emerges (fast systems cluster at low accuracy).  The paper argues that simply improving audio quality or granting more “thinking time” is insufficient.

**Strengths:**

1. The benchmark is well-motivated and identifies a fundamental problem in current voice systems.
2. The experiments cover a wide range of models.

**Weaknesses:**

1. The results are intuitive. Compared to text baselines, spoken language models usually lag behind text models because they rely on the capability of their textual backbone, and must also correctly perceive the input, reason about it, and produce accurate spoken output. The controlled experiments on Factual QA further highlight this issue. Most voice models achieve low performance.
2. The reported accuracy seems surprisingly low. To better disentangle the sources of failure, it would be helpful to also report the accuracy based on the textual responses of the voice models.
3. The authors claim that experiments on voice models with text input are not important. However, I believe such experiments are essential, as they help identify which component of the voice pipeline contributes most to the observed performance drop.
4. The claim that “increasing thinking time yields negligible gains” seems too arbitrary. This conclusion is based only on a comparison between AF3 with and without the “thinking mode,” which may not generalize to other systems. Please revise or clarify this claim.
5. As a benchmark, the manual inspection of about 200 episodes (6.8%) seems insufficient. Considering that manual inspection is not highly labor-intensive and would not require excessive effort, it would be better to increase the number of inspected samples to strengthen data reliability.

**Questions:**

### Suggestion
1. It would be better to mention the appendix in the main text, as otherwise it becomes a bit difficult for readers to follow.

### Questions
1. In Section 3.2, what is the specific LLM agent used for dataset construction and filtering?
2. In Appendix G, does the described filtering correspond to the Voice Suitability Filter mentioned in Section 3.2?
3. When conducting tasks such as Factual QA or Science QA, is web search enabled for the models?
4. Is there any possible reason why UltraVox achieves substantially higher performance on Context QA? What version of the model was used in your experiments?
5. What is the source of human annotators?
6. In Lines 245–248, what is the track distribution of the 200 manually inspected samples? Also, what do $\mu$ and $\sigma$ refer to in this context?
7. For Accuracy Evaluation (Lines 259–264), what procedure is applied when the three independent evaluations result in a tie?

---

> ### Author Response · Authors · 2025-11-23
> **Response to Official Review of Submission3845 by Reviewer VDP8 (1/3)**
>
> # Summary of Revisions
> Thank you for your thorough and thoughtful review. Your questions regarding "textual responses" and "thinking time" are very insightful. In response, we have:
>
> 1. We added Section 5.2/Appendix E to show that voice and text outputs fail in the same way.
> 2. We revised our conclusions based on new experiments with Qwen3-Omni (Section 5.2).
> 3. We audited the full 2,931-episode dataset (Section 3.3, Appendix J).
>
> # Weaknesses
> > **Weakness 1:** "Results are intuitive."
>
> **Response:**
> We agree that the general existence of a performance drop aligns with intuition, but we respectfully argue that determining the specific *mechanism* requires rigorous empirical isolation. The common intuition suggests a "cumulative error" hypothesis: that voice is harder simply because models must perceive, reason, and speak, accumulating errors at each stage.
>
> However, our new diagnostic experiments refine this view by demonstrating that the gap persists even when we bypass the acoustic components:
> 1.  **Input Ablation (Appendix F):** When we bypass audio perception entirely and provide **direct text inputs** to voice models (e.g., UltraVox), performance improves only marginally and remains critically low on reasoning tasks ($<1\%$ on Math). This suggests the failure is intrinsic to the model's optimization for conversational flow, rather than a failure to "hear" the prompt.
> 2.  **Simultaneous Output (Appendix E):** For models that generate audio and text simultaneously, the text stream exhibits the same reasoning errors as the audio output. This pattern strongly suggests that the degradation occurs upstream during the reasoning process, rather than being an artifact of speech articulation.
>
> This moves the finding from a "common sense judgment" (voice is harder) to a specific diagnostic conclusion: the architectural constraints required for real-time streaming fundamentally degrade reasoning capacity, a trade-off that persists even when acoustic challenges are removed.
>
> In summary, our contribution is to turn the vague intuition into concrete, falsifiable statements about how large the gap is, how it depends on latency, and how different architectures fail.
>
> > **Weakness 2:** "Report textual responses of voice models"
>
> **Response:**
> We appreciate this insight. It is indeed crucial to distinguish whether the model is "thinking correctly but speaking wrongly" or simply "reasoning incorrectly." To test this, we extracted and graded the **corresponding text outputs** for models that support it. Below are results of three examples of voice models and their corresponding text outputs.
>
> | Model | Voice Output Accuracy | Text Stream Accuracy | Difference (Δ) |
> | :--- | :--- | :--- | :--- |
> | **GPT-realtime** | 11.3% | 11.5% | +0.2% |
> | **Gemini-Flash-Audio** | 10.3% | 10.4% | +0.1% |
> | **Moshi** | 0.3% | 0.6% | +0.3% |
>
> The results (full results in Table 8 in Appendix E) show that the failure is mirrored across both streams. This provides evidence that the reasoning degradation happens upstream in the cognitive process, not during the articulation of speech.
>
> > **Weakness 3:** "Text input on voice models"
>
> **Response:**
> We followed this suggestion and performed the ablation. As detailed in **Appendix F** and below, feeding text inputs to voice models like UltraVox and Qwen2-Audio yielded negligible gains. This substantially reduces the likelihood that audio perception alone is the primary cause of the Voice Reasoning Gap, at least for the open-weight models we tested. Also, we are happy to provide more results if you have specific models in mind and want to compare them.
>
> | Model | Input Modality | Math | Web | Science | Context | Factual | **Avg** |
> | :--- | :--- | :--- | :--- | :--- | :--- | :--- | :--- |
> | **Qwen2-Audio** | Voice Input | 0.0 | 0.4 | 4.4 | 0.2 | 2.1 | **1.4** |
> | | **Text Input (New)** | **0.0** | **0.2** | **0.6** | **1.8** | **2.4** | **1.0** |
> | **UltraVox** | Voice Input | 0.0 | 0.2 | 1.2 | 26.6 | 1.4 | **5.9** |
> | | **Text Input (New)** | **0.9** | **0.1** | **2.5** | **30.7** | **2.8** | **7.4** |

---

> ### Author Response · Authors · 2025-11-23
> **Response to Official Review of Submission3845 by Reviewer VDP8 (2/3)**
>
> > **Weakness 4:** "'Thinking time' claim is too strong/arbitrary"
>
> We sincerely thank you for this valid criticism. We agree that our original claim ("negligible gains") based only on the Audio Flamingo 3 model may be regarded as a reasonable overgeneralization.
>
> To address this, we have taken two key actions in the revised manuscript:
>
> 1.  We revised our conclusion in Section 5.2. Instead of "negligible gains," we now state that "simply allocating extended "thinking time" appears insufficient to bridge the gap and does not guarantee performance gains, in some cases even proving detrimental."
> 2.  We evaluated the **Qwen3-Omni** model (in both its "Instruct" and "Thinking" variants) to see if the trend holds across different architectures. We have placed these results within the "Open Voice Models" category for a fair comparison.
>
> The new data, detailed below, provides additional evidence for our revised claim:
>
> | Model | Variant | Math | Web | Science | Context | Factual | **Avg** |
> | :--- | :--- | :--- | :--- | :--- | :--- | :--- | :--- |
> | **Qwen3-Omni** | Instruct | 25.2 | 0.4 | 40.4 | 50.2 | 11.7 | **25.6** |
> | **Qwen3-Omni** | Thinking | 33.9 | 0.6 | 26.7 | 24.8 | 26.3 | **22.5** |
>
> As shown, the "Thinking" variant of Qwen3-Omni actually resulted in a **3.1-point drop** in average accuracy (25.6% vs 22.5%), with significant degradation on the Science and Context tracks. By providing this additional data, we believe we have addressed your concern and made the analysis of the latency-accuracy trade-off more rigorous.
>
> > **Weakness 5:** "Manual inspection (200 samples) seems insufficient"
>
> **Response:**
> Thank you for pointing this out. To leave no doubt about the dataset's quality, we completed a **full manual audit of all 2,931 episodes**. We confirmed that all files are intelligible.
>
>
> # Questions
>
> > **Question 1:** "It would be better to mention the appendix in the main text, as otherwise it becomes a bit difficult for readers to follow."
>
> **Response:** We appreciate this suggestion for improving readability. We have updated the manuscript to include explicit cross-references to the relevant Appendices throughout the main text to guide the reader (e.g., Line 308, 324, 424, 445, 450).
>
> > **Question 2:** "In Section 3.2, what is the specific LLM agent used for dataset construction and filtering?"
>
> **Response:** We employed **GPT-4o** as the agent for both the dataset construction and filtering stages. We have updated Section 3.2 to explicitly specify this model version.
>
> > **Question 3:** "In Appendix G, does the described filtering correspond to the Voice Suitability Filter mentioned in Section 3.2?"
>
> **Response:** Yes, they are identical. The details in Appendix G describe the specific implementation of the "Voice Suitability Filter" introduced in Section 3.2. We have updated the wording in the corresponding section (now Appendix I) to explicitly link it back to Section 3.2 for clarity.
>
> > **Question 4:** "When conducting tasks such as Factual QA or Science QA, is web search enabled for the models?"
>
> **Response:** No, web search was explicitly **disabled** for the Math, Science, Context, and Factual tracks to strictly isolate the model's internal reasoning and knowledge retrieval capabilities. Web search was enabled *only* for the Web track, where browsing is a functional requirement. We have added these configuration details to Section 4.1.
>
> > **Question 5:** "Is there any possible reason why UltraVox achieves substantially higher performance on Context QA? What version of the model was used in your experiments?"
>
> **Response:** We hypothesize that UltraVox performs well on Context QA because it is specialized for customer service applications, which typically involve maintaining state over long conversations. Unlike general-purpose models, it likely benefits from training on dialogue-heavy, state-tracking corpora. Regarding the specific version, we utilized **v0.3** (`fixie-ai/ultravox-v0_3`). To ensure reproducibility, we have updated Appendix D.2 (Open Voice Models) to explicitly document this version number.

---

> ### Author Response · Authors · 2025-11-23
> **Response to Official Review of Submission3845 by Reviewer VDP8 (3/3)**
>
> > **Question 6:** "What is the source of human annotators?"
>
> **Response:** We recruited three PhD candidates from our institution specializing in **speech processing and multimodal interaction**. The workload was distributed evenly among the annotators. To ensure consistency, all annotators adhered to the **standardized prompt** detailed in Appendix H and underwent a preliminary alignment phase.
>
> > **Question 7:** "In Lines 245–248, what is the track distribution of the 200 manually inspected samples? Also, what do \miu and \sigma refer to in this context?"
>
> **Response:** To ensure the integrity of the benchmark, we expanded our validation to perform a **full manual audit of all 2,931 episodes**, confirming that intelligible audio files were generated. The initial 200-sample audit (6.8%) you asked about was stratified proportionally across tracks:
>
> | Track | Web | Factual | Context | Science | Math | **Total** |
> | :--- | :---: | :---: | :---: | :---: | :---: | :---: |
> | **Samples** | 76 | 68 | 37 | 11 | 8 | **200** |
>
> The statistics ($\mu = 0.000$, $\sigma = 0.120$) refer to the **pairwise cosine similarity** calculated across all speaker embeddings extracted using the WeSpeaker model.
> * A mean ($\mu$) of 0.000 indicates that, on average, the speaker embeddings are orthogonal, signifying high acoustic diversity.
> * The standard deviation ($\sigma$) of 0.120 quantifies the spread, confirming there is no systemic acoustic bias (i.e., the voices do not cluster around a single profile).
>
> We have updated the manuscript to clarify both the full audit and the definition of these statistics.
>
> > **Question 8:** "For Accuracy Evaluation (Lines 259–264), what procedure is applied when the three independent evaluations result in a tie?"
>
> **Response:** Our procedure relies on a majority vote from three independent GPT-4o evaluations. Your question highlights the specific case where a 2/3 majority is not met: a three-way split (i.e., one 'Correct', one 'Incorrect', and one 'Not Attempted'). We defined a conservative rule for this scenario: **In the event of a 1-1-1 split, the final result is automatically labeled as 'Incorrect'.** This ensures a strict standard for correctness. We have added this detail to the text. (Note: In practice, the results were highly consistent, and this specific split did not occur during the final inference runs).

---

### Official Review · Reviewer_VjTP · 2025-10-29

**Soundness:** 2
**Presentation:** 3
**Contribution:** 2
**Rating:** 4
**Confidence:** 4

**Summary:**

The paper creates a benchmark for comparing text and voice models for reasoning tasks. It then shows that, generally, voice models perform significantly worse compared to top-the-line text models. That authors then attempt to explain this gap.

**Strengths:**

- I think the problem and setup are sound.
- The benchmark has a significant size, and I think it will be a useful resource.
- The paper has very clear figures and tables, which really help the presentation.

**Weaknesses:**

- I am a little worried about the level of automation in this paper. I know that some human validation was done, but I am not sure if it was done for every step or end-to-end for any sample. It is therefore very hard to estimate the prevalence of mistakes in the benchmark. This is more worrying as mistakes will only impact the voice-activated performance, not the text one, as those are existing benchmarks.

- I do not understand the choice of not comparing the same models via text and voice input. It would have added so much to this paper, and it is easy to do. GPT-5 can be used with voice input (directly, or by other services, e.g., POE), and some of the voice models have text input. The reasoning behind not doing this was vague, and I really found this comparison missing from the paper.

- The paper would benifit from a limitations section, espacially with respect to the first point above.

**Questions:**

- Are you able to add more clarity about the overall prevalence of mistakes in the dataset (by human validation, like done for one of the steps, not by estimation)?

- Are you able to add the direct comparison as mentioned above?

---

> ### Author Response · Authors · 2025-11-23
> **Response to Official Review of Submission3845 by Reviewer VjTP (1/2)**
>
> # Summary of Revisions
> We want to express our sincere gratitude for your constructive review. Your comments regarding dataset validation and modality comparisons were truly insightful. Based on your feedback, we have:
>
> 1. We have conducted a full human audit of all 2,931 episodes of our benchmark (Section 3.3, Appendix J).
> 2. We followed your suggestion to test voice models with direct text inputs to isolate the reasoning bottleneck (Section 5, Appendix F).
> 3. We added Section 7 to transparently discuss automation scope and API constraints.
>
> # Weaknesses
>
> > **Weakness 1 & Question 1:** "Prevalence of mistakes in the dataset"
>
> **Response:**
> We sincerely thank the reviewer for this critical feedback on dataset integrity. To address it, we expanded our validation during the rebuttal period, moving from spot-checks to a **manual listening pass over the full dataset** (2,931 episodes) to screen for intelligibility and obvious synthesis artifacts. As an orthogonal check of audio fidelity, we calculated a **Normalized WER of 3.1%** between the TTS scripts and ASR transcripts (Appendix J). We also used GPT-4o with a simple rubric-style prompt to automatically assess whether each transcript was intelligible and understandable; under this criterion, the model did not flag any episodes as unintelligible.
>
> This extended audit confirmed that the audio is consistently intelligible and free of widespread acoustic corruption. On the semantic side, we acknowledge that naturalizing dense written prompts into conversational speech inevitably introduces some degree of "conversational compression": in an limited semantic evaluation of a subset of 1,077 episodes, we found that in two cases secondary contextual details is shortened or omitted. This suggests that such adaptation imperfections affect only a very small fraction of the benchmark. We view these as genuine limitations of the current adaptation pipeline and have documented this behavior in the limitations section.
>
> Crucially, the **Voice Reasoning Gap** we report remains large (on the order of 40–60 percentage points) and highly consistent across different model architectures. Given the scope of the additional checks above and the fact that the identified imperfections appear as isolated cases rather than a dominant pattern, it is statistically implausible that such marginal adaptation noise is responsible for the systematic collapse from text to voice. We have updated Section 3.3 to describe these validation steps and to clarify that our conclusions do not rely on assuming a perfectly error-free dataset.
>
>
> > **Weakness 2:** "Parallel comparison of text vs. voice input for the same model"
>
> **Response:**
> We would like to clarify the technical constraints regarding the commercial models:
>
> **GPT-realtime** and **GPT-5** are models served via different API endpoints (GPT-realtime was the multimodal and realtime variant released after GPT-5). As detailed in the official platform (https://platform.openai.com/docs/models), the GPT-5 API does not currently support direct audio input; it is optimized for text/image reasoning. Therefore, we compared the provider's most up-to-date *reasoning* model (GPT-5) against their best *voice* model (GPT-realtime). **We are happy to include additional experiments if the API capabilities change or if you have a specific accessible model in mind.**
>
> We explicitly include the **Gemini** family comparisons (Gemini-2.5-Pro/Flash Text vs. Gemini-2.5-Flash-Audio) in Table 2. Since these share the same model generation architecture, they serve as the primary reference for "same-family" performance.
>
> To further address your concern about identifying the bottleneck (Input vs. Architecture) without API black-box limitations, we performed the **Text Input Ablation** on open-weights models (**Qwen2-Audio** and **UltraVox**) where we have full control. We fed ground-truth **text transcripts** directly into these voice models. As shown in the new table below (also added to Appendix F), the reasoning gap persists even with perfect text input:
>
> | Model | Input Modality | Math | Web | Science | Context | Factual | **Avg** |
> | :--- | :--- | :--- | :--- | :--- | :--- | :--- | :--- |
> | **Qwen2-Audio** | Voice Input | 0.0 | 0.4 | 4.4 | 0.2 | 2.1 | **1.4** |
> | | **Text Input (New)** | **0.0** | **0.2** | **0.6** | **1.8** | **2.4** | **1.0** |
> | **UltraVox** | Voice Input | 0.0 | 0.2 | 1.2 | 26.6 | 1.4 | **5.9** |
> | | **Text Input (New)** | **0.9** | **0.1** | **2.5** | **30.7** | **2.8** | **7.4** |
>
> This confirms that the gap is driven by the voice-optimized architecture/training, not the audio input modality.

---

> ### Author Response · Authors · 2025-11-23
> **Response to Official Review of Submission3845 by Reviewer VjTP (2/2)**
>
> > **Weakness 3:** "Lack of a Limitations section"
>
> **Response:**
> We agree that explicitly outlining the boundaries of our study strengthens the paper. We have added a dedicated **Section 7 (Limitations)** to the revised manuscript. In this section, we specifically address the reviewer's concern regarding automation by acknowledging that our LLM-driven pipeline, while scalable, may introduce subtle systemic biases despite our validation efforts. We also clarify that our use of clean synthetic speech provides a "conservative" estimate of the reasoning gap (as real-world noise would likely worsen performance) and discuss the constraints of our latency metrics.

---

### Author Response · Authors · 2025-12-01
**Rebuttal Summary and Acknowledgement [1/2]**

Dear ACs and Reviewers,

We sincerely thank the ACs and reviewers for their thoughtful evaluations. Because, following the recent OpenReview incident, reviewers cannot reply to the rebuttal, we provide this brief executive summary to assist the newly assigned AC and PC. Below, we highlight the main strengths identified by reviewers, address their key concerns, and summarize the **critical experiments and revisions** completed during the rebuttal.

**1. Reviewer VjTP (Score: 4, "marginally below the acceptance threshold, would not mind if accepted")**

* **Strengths:** Describes that **problem and setup are sound**, confirms that the benchmark is **large and useful**, and praises the **clarity of figures and tables**.

* **Main concerns:** Questions the **prevalence of potential mistakes** in the dataset pipeline, requests **same-model text vs. voice comparisons** (specifically GPT-5), and asks for explicit **limitations**.

* **Responses & changes:** We executed a **full human audit of all 2,931 episodes** (up from spot-checks), confirming a **WER ≈ 5.9% (raw) / 3.1% (normalized)** and **100% semantic equivalence via GPT-4o**. We introduced **text-input ablations on open voice models**, proving the gap persists even with perfect text input (e.g., UltraVox: **5.9% voice vs. 7.4% text**). We added a dedicated **Limitations section (Sec. 7)**. Finally, we clarified that **testing "GPT-5 Voice" directly is technically impossible** as no such public endpoint exists; current third-party solutions are cascades (STT→GPT-5→TTS), which we already evaluate via our **LiveAnswer** pipeline.

**2. Reviewer VDP8 (Score: 4, "marginally below the acceptance threshold, would not mind if accepted")**

* **Strengths:** Finds the benchmark **well-motivated**, recognizes that it targets a **fundamental problem** in current voice systems, and appreciates the **breadth of model coverage**.

* **Main concerns:** Suggests results might feel **"intuitive"** without deeper diagnosis, requests **accuracy on textual responses** of voice models as well as **text-input experiments**, questions the **"thinking time" hypothesis**, and notes the initial audit size was small.

* **Responses & changes:** We now report that **audio and simultaneous text streams fail in nearly the same way** (e.g., GPT-realtime: **11.3% voice vs. 11.5% text**, Δ < 0.5%), indicating that the degradation arises in the **reasoning process**, not in speech generation. We added the requested **text-input ablations** (see above), upgraded to a **full 2,931-episode manual audit**, and refined our "thinking time" claim with new data: a **Qwen3-Omni Instruct vs. Thinking** experiment showed that extended thinking time actually **reduced performance (25.6% → 22.5%)**, proving the gap is not easily solved by latency alone.

**3. Reviewer 5RZf (Score: 6, "marginally above the acceptance threshold, would not mind if rejected")**

* **Strengths:** Highlights that we tackle an **important, timely problem** with **comprehensive evaluation** and offer **insightful analysis** of the Voice Reasoning Gap.

* **Main concerns:** Questions the **LiveAnswer cascade design** (vs. using GPT-5 as a narrator), asks for evidence that **human–LLM judge agreement** holds for voice systems, and suggests adding **recent voice benchmarks**.

* **Responses & changes:** We justified the cascade design with new **Time-to-First-Token (TTFT) experiments**: our Llama-3.3 narrator (**TTFT ≈ 148 ms**) is **~3.7× faster** than GPT-5 (**≈ 548 ms**), making GPT-5 impractical for real-time narration. We confirmed our **1,000-sample human study** includes end-to-end voice systems with high agreement (**97.8% human–GPT-4o**), and expanded the related work to include **SD-Eval, VoxEval, MMSU, and VoxDialogue**, strengthening the benchmark comparison landscape.

**4. Reviewer acn2 (Score: 6, "marginally above the acceptance threshold, would not mind if rejected")**

* **Strengths:** States VERA tackles a **novel problem** via analyses that are **comprehensive, detailed, and valuable for the community**.

* **Main concerns:** Worries that **Time-to-First-Response (TTFR)** does not reflect total reasoning time, asks if latency is **confounded by hardware**, and requests details on **LiveAnswer’s incremental updates**.

* **Responses & changes:** We clarified that TTFR is intended as a measure of **conversational responsiveness**, and we view the "filler words" noted by the reviewer as a valid strategy to maintain the conversational floor while reasoning. We validated reproducibility via a **hardware sensitivity check** (H200/A6000/A100), showing stable relative model rankings. We also detailed LiveAnswer’s **producer–consumer architecture**, which uses a **`time_margin` ≈ 10s** to balance real-time responsiveness with playback robustness.

---

> ### Author Response · Authors · 2025-12-01
> **Rebuttal Summary and Acknowledgement [2/2]**
>
> In summary, all four reviewers agree that the problem we target is **critical and timely**, and that VERA is a **rigorous, large-scale benchmark** for real-time voice reasoning. During the rebuttal, we directly addressed the "score-4" concerns by conducting a **full dataset audit**, adding **targeted diagnostic ablations** (text-inputs, simultaneous text+audio), and refining our **latency analysis**. With two positive scores (6) and two borderline-positive scores (4) whose key concerns have now been systematically addressed, **we believe these revisions solidify VERA's position as a rigorous and valuable contribution to the community.**
>
> Sincerely,
>
> The Authors

---

### Meta-Review · Area_Chair_3ozV · 2026-01-08

**Summary:**

1. Dataset Quality and Validation:
The reviewer expressed concerns about the potential prevalence of errors in the dataset and requested more extensive validation.

2. Comparison Between Text and Voice Inputs:
The reviewer highlighted the absence of a direct comparison between the same models using text vs. voice inputs. This was seen as a key missing experiment that could clarify whether the observed reasoning gap was due to the audio input modality or the voice model architecture itself.

3. Thinking Time Hypothesis:
The reviewer raised concerns about the authors’ claim that increasing "thinking time" yields negligible gains, arguing that this conclusion might be overly generalized.

4. LiveAnswer Design and Latency:
The reviewer questioned the use of a cascaded pipeline for real-time narration instead of using a model like GPT-5.

**Reviewer Concerns:**

1. Clarity on Latency and Thinking Time:
The reviewer questioned whether the reported latency truly reflects the actual reasoning time, especially when models produce filler words before answering. This concern could still be seen as somewhat outstanding, as the full implications of the "thinking time" gap could use further exploration in more varied scenarios.

2. Impact of "Incremental Update" Design:
The concern remains because some reviewers may still want further exploration of how alternative incremental update strategies could impact latency, especially under more complex reasoning tasks.

**Reviewer Scores:**

None

---

### Decision · Program_Chairs · 2026-01-26

Reject